# Phylogenetic Analysis Reveals Four New Species of *Otidea* from China

**DOI:** 10.3390/biology11060866

**Published:** 2022-06-06

**Authors:** Yu-Yan Xu, Ming-Qi Zhu, Ning Mao, Li Fan, Xiao-Ye Shen

**Affiliations:** 1College of Life Science, Capital Normal University, Xisanhuanbeilu 105, Haidian, Beijing 100048, China; 2190801004@cnu.edu.cn (Y.-Y.X.); 2210801013@cnu.edu.cn (N.M.); 2College of Plant Protection, Northwest A & F University, Xinonglu 22, Yangling, Shaanxi 712100, China; zhumingqi@nwsuaf.edu.cn

**Keywords:** Ascomycota, Pyronemataceae, phylogeny, taxonomy

## Abstract

**Simple Summary:**

*Otidea* is a remarkable genus of ascomycetes. Members of the genus are diverse in ascomatal form, with split to entire, sessile to stipitate, cupulate to ear-shaped epigeous apothecia, as well as closed, solid ascomata in hypogeous taxa. They are distributed in the temperate to arctic-alpine regions of the northern hemisphere. China, located in the northern hemisphere, has a vast temperate zone, and a lot of new species of *Otidea* have been proposed recently. In this study, some specimens deposited in Chinese fungus herbaria and one newly collected *Otidea* specimen from northern China were studied using morphological and phylogenetic methods. The results indicate the presence of nine phylogenetic species of *Otidea* within the sample, four of them are described as new species, namely *O. bomiensis*, *O. gongnaisiensis*, *O. hanzhongensis*, and *O. shennongjiana*. Recognizing these new species will increase knowledge of species resources of *Otidea* in China.

**Abstract:**

The emergence of molecular systematics has greatly helped researchers to identify fungal species. China has abundant *Otidea* species resources, and a number of new species of *Otidea* have been recently proposed. However, many old specimens in herbaria are mainly identified by morphology rather than molecular methods. In this study, 11 specimens deposited in Chinese herbaria and one newly collected *Otidea* species from northern China were identified based on morphological and phylogenetic analyses. Four gene fragments (ITS, LSU, *rpb2*, and *tef1-α*) were used to elucidate the phylogenetic relationships of species within *Otidea.* A total of nine phylogenetic species are recognized, of which four are described as new species, namely *O. bomiensis*, *O. gongnaisiensis*, *O. hanzhongensis*, and *O. shennongjiana*. Among the known species were *O. aspera* and *O. sinensis*.

## 1. Introduction

The genus *Otidea* (Pers.) Bonord. (*Pyronemataceae*, *Pezizales*) was established by Bonorden, who elevated *Peziza* subgenus *Otidea* to generic rank, and typified it by *Otidea onotica* (Pers.) Fuckel [1,2,3]. *Otidea* is a monophyletic but morphologically diverse genus. Members of the genus exhibit a wide range of ascomatal forms, with split to entire, sessile to stipitate, cupulate to ear-shaped epigeous apothecia, as well as closed, solid ascomata in hypogeous taxa, and are mainly distributed in temperate zones of Europe, North America, and Asia in the northern hemisphere, with a few from the arctic-alpine regions. Species of *Otidea* are considered to be ectomycorrhizal and thus play important roles in forest ecosystems [4,5,6,7,8,9].

Hansen and Olariaga [7] used multilocus phylogenies (LSU, *rpb2*, *tef1-α*) to elucidate species relationships within *Otidea*. Olariaga et al. [8] simultaneously published a monograph of the genus based on relatively extensive taxon sampling and introduced new morphological and histochemical features for species identification. These works laid the foundation for subsequent research of *Otidea*. In recent years, a number of new species have been discovered both in Europe and Asia [9,10,11,12,13,14].

During a previous investigation on fungal resources of northern China, we collected many fresh apothecia of the genus *Otidea* and identified seven new species that were described and illustrated [15]. More recently, following the examination of 11 herbarium specimens (HMAS and HMUABO), four additional *Otidea* species were discovered that appeared to be undescribed. In this study, we report on the detailed analyses of these Chinese specimens using morphological and phylogenetic methods. Our goals are (i) to evaluate the species identity of herbarium specimens and one freshly collected specimen, (ii) to infer the phylogenetic position of species within *Otidea* based on multigene sequences (ITS, LSU, *tef1-α, rpb2)*, and (iii) to describe and illustrate the four new species.

## 2. Materials and Methods

### 2.1. Morphological Studies

Specimens were studied from HMAS (Herbarium Mycologicum Academiae Sinicae, Institute of Microbiology, Chinese Academy of Sciences) and HMUABO (Fungal Herbarium of Northwest A & F University, Yangling, China). Macroscopic characters were recorded from fresh specimens. Microscopic characters were observed in 10% KOH, Congo Red, and Melzer’s reagent [16]. The form ‘(a-)b-c(-d)’ is used to represent the dimensions of ascospores. The range b–c contains a minimum of 90% of the measured values. The extreme values, i.e., a and d, are given in parentheses. L_m_ and W_m_ are used to represent the average ascospore length and width. Q means length/width ratio; Q_m_ means the average Q. ‘n’ means the number of populations.

### 2.2. DNA Extraction, PCR Amplification, Sequencing

Portions of dried specimens were crushed, and then DNA was extracted using the method of CTAB [17]. ITS1f/ITS4 [17,18] were used for the internal transcribed spacers of the nuc rDNA (ITS), LR0R/LR5 [19] for the nuc rDNA 28S subunit (LSU), RPB2-Otidea6F/RPB2-Otidea7R and fRPB2-7cF/fRPB2-11aR [7] for the RNA polymerase II second largest subunit (*rpb2*), and EF1-983F/EF1-2218R [20] for the translation elongation factor 1-alpha (*tef1-α*), respectively. PCR reactions and procedures were as described in Xu et al. [15]. PCR products were sent to Sangon Biotech (Shanghai) Co., Ltd. (Beijing, China) for purification and sequencing. The accession numbers of newly obtained sequences used for phylogenetic analyses are provided in Appendix A.

### 2.3. Sequence Alignment and Phylogenetic Analyses

Three datasets were assembled for this study. Dataset I (LSU-*tef1-α*-*rpb2**)* contained our collections and the backbone species in all phyloclades of *Otidea*, which was used to infer the phylogenetic status of our collections at the phyloclade level in the genus *Otidea*. *Warcupia terrestris* Paden & J.V. Cameron and *Monascella botryosa* Guarro & Arx were used as outgroups. Datasets II and III (ITS-LSU) corresponded to the *Otidea formicarum* phyloclade and the *O. alutacea* phyloclade, respectively, and contained the representative species of their own and all available sequences of Chinese *Otidea* species (including our collections) separately classified in these two clades. These two datasets were used to evaluate the relationship between the four new species and their close relatives. *Otidea cantharella* (Fr.) Quél. and *O. platyspora* Nannf. were selected as outgroups for Datasets II and III, respectively.

The sequences (ITS, LSU, *tef1-α**,* and *rpb2*) were independently aligned in an online version of MAFFT 7.110 using default parameters [21] and manually modified where necessary in Se-Al 2.03a. [22]. Ambiguous aligned regions for each sequence were detected and excluded through Gblocks 0.91b [23] using default options with default parameters. To check for the degree of congruence among different genes, phylogenetic analyses were first conducted separately for each gene and later for the genes combined. These genes were then concatenated using SequenceMatrix v1.7.8 [24], and the alignment files of the three datasets were provided in Appendix A. Maximum likelihood (ML) and Bayesian inference (BI) were used to infer phylogenetic analyses.

ML analysis was performed in RAxML [25] by running 1000 bootstrap replicates with all parameters at default settings using the GTRGAMMAI model. BI analysis was performed in MrBayes [26] based on the best substitution models (GTR+I+G for ITS, LSU, and *rpb2*; SYM+I+G for *tef1-α*) determined by MrModeltest 2.3 [27]. Two independent runs of four chains were conducted: 1,000,000 for the LSU/*tef1-α*/*rpb2* dataset, 395,000 for the *O. formicarum* dataset, and 615,000 for the *O. alutacea* dataset. Markov Chain Monte Carlo generations were conducted using the default settings and sampled every 100 generations. The average standard deviations of split frequency (ASDSF) values were far lower than 0.01 at the end of the runs. Trees were sampled every 100 generations after burn-in, and 50% of the majority-rule consensus trees were constructed. Maximum likelihood bootstrap support (BS) ≥ 50% and Bayesian posterior probability (PP) ≥ 0.95 were shown on the nodes [28,29]. Trees were visualized using TreeView [30].

## 3. Results

### 3.1. Phylogenetic Analyses

The combined LSU/*tef1-α*/*rpb2* dataset contained 360 sequences from 146 taxa, including 39 newly obtained sequences in this study. The length of the aligned dataset was 3547 bp (802 bp for LSU, 969 bp for *tef1-α*, 1776 bp for *rpb2*). The topologies of ML and BI phylogenetic trees were similar, so only the ML tree is shown (Figure 1). The *Otidea* species formed a monophyletic lineage with high support values. Our *Otidea* collections were nested in four clades, i.e., *O. bufonia-onotica* clade, *O.*
*concinna* clade, *O. formicarum* clade, and *O.*
*alutacea* clade, and a total of nine phylogenetic species were recognized from our Chinese collections. In the *O. bufonia-onotica* clade, one species was recognized. Although this species contained one collection, HMAS 85660, and probably represented an undescribed species, we could not describe it due to the poor condition of its ascomata (see discussion). Here, we temporally named it *Otidea* sp.’c’. In the *O.*
*concinna* clade, two collections were placed in a well-supported clade, represented by a known species, *O.*
*sinensis* J.Z. Cao & L. Fan. In the *O. formicarum* clade, two species were recognized. They were described as new species *O.*
*gongnaisiensis* and *O.*
*shennongjiana,* respectively. In the *O. alutacea* clade, five species were recognized. Two of them were described as new species, *O.*
*bomiensis* and *O.*
*hanzhongensis,* in this paper. One (HSA 251) is corresponding to the known species *O.*
*aspera* L. Fan & Y.Y. Xu. The remaining two phylogenetic species (clades 6 and 7) were not treated taxonomically here due to the poor condition of Chinese collections included in the two clades (see discussion). Further, we could not examine the Swedish specimen (C-F-48045).

The O. formicarum dataset contained 42 sequences from 21 taxa, including four newly obtained sequences and four sequences from outgroups (*O. cantharella*). The length of the aligned dataset was 1416 bp (582 bp for ITS, and 834 bp for LSU). The topologies of ML and BI phylogenetic trees were similar, so only the ML tree is shown (Figure 2). Similar to the three-gene phylogeny result, the collections from China formed two independent branches with high supports, representing two new species, *O. gongnaisiensis* and *O. shennongjiana*. *Otidea gongnaisiensis* was further grouped together with *O. formicarum* with moderate support (BS = 71%, PP = 0.99), and *O. shennongjiana* was sister to *Otidea* sp. ‘b’ (KH.09.79) with high bootstrap support.

The *O**. alutacea* dataset contained 81 sequences from 41 taxa, including ten newly obtained sequences and four sequences from outgroups (*O. platyspora*). The length of the aligned dataset was 1357 bp (572 bp for ITS, and 785 bp for LSU). The topologies of ML and BI phylogenetic trees were similar, so only the ML tree is shown (Figure 3). Similar to the three-gene phylogram result, the Chinese collections were placed in five well-supported clades. Of them, one clade is represented by a known species (*O. aspera*), and two are new species (*O. bomiensis* and *O. hanzhongensis*). The remaining two clades (clades 6 and 7) are two distinct species, but we are not able to treat these two taxa taxonomically (see discussion).

### 3.2. Taxonomy

Based on our phylogenies and morphological data, four new species and two known species of *Otidea* from China were described and illustrated here.

***Otidea******bomiensis*** L. Fan & Y.Y. Xu, sp. nov. (Figure 4)

MycoBank: MB844136

Etymology: *bomiensis*, referring to the locality where the type specimen was collected.

Holotype: China. Tibet Autonomous Region, Nyingchi City, Bomi County, Zhamu Town, Daxing Village, alt. 2900–3000 m, on soil in mixed forest, 28 August 1983, X.L. Mao M1430 (HMAS 52743).

Saprobic on soil. Apothecia solitary or gregarious, 10–40 mm high, 5–25 mm wide, broadly ear-shaped to shallowly or deeply cup-shaped, split, stipitate. Hymenium light brown to brown when fresh, dark brown when dry, margin with purpure tones, subsmooth. Receptacle surface pale yellow when fresh, slightly hygrophanous, pale yellow brown to yellowish brown when dry, furfuraceous to finely warty. Stipe 7–15 × 5–7 µm. Basal tomentum and mycelium white. Apothecial section 700–1000 µm thick. Ectal excipulum of *textura angularis*, 80–120 µm thick, cells thin-walled to slightly thick-walled, pale brown, 12–30 × 10–21 µm. Medullary excipulum of *textura intricata*, 300–550 µm thick, hyphae 3–9 µm wide, thin-walled, septate, hyaline to light brown. Subhymenium c. 40–60 µm thick, visible as a brown zone, of densely arranged cylindrical to swollen cells. Paraphyses septate, bent to curved, a few straight, of uniform width or slightly enlarged at the apices to 3–4.5 µm wide, without or with a low notch. Asci 150–200 × 10–14 µm, 8-spored, unitunicate, operculate, cylindrical, hyaline, non-amyloid, long pedicellate, arising from croziers, ascospores released from an eccentric split at the apical apex. Ascospores overlapping uniseriate, ellipsoid to oblong ellipsoid, hyaline, with one to two large guttules, smooth, (14.5–)15–16.8(–17.2) × (6.5–)7.1–7.9(–8.3) µm (L_m_ × W_m_ = 15.9 × 7.5 µm, Q = 1.9–2.2, Q_m_ = 2, n = 50). Receptacle surface with broad conical warts, 30–50 µm high, formed by short, fasciculate, hyphoid hairs, of 4–6 subglobose to elongated cells, constricted at septa, 5–9 µm wide. Resinous exudates absent to scarce. Basal mycelium of interwoven, 3–6 µm wide, septate, hyaline to pale brown hyphae, unchanged in KOH, smooth.

Additional specimens examined: China. Sichuan Province, Xiangcheng County, Daxueshan, alt. 4300 m, on soil under *Rhododendron* sp., 25 July 1998, Z. Wang, WZ185 (HMAS 75178).

Notes: *Otidea bomiensis* is characterized by the stipitate, light brown to brown hymenium, pale yellow receptacle surface, the lack of resinous exudates on the ectal excipulum and basal mycelium, ellipsoid to oblong ellipsoid ascospores and bent to curved paraphyses. The hymenium color of *O. adorniae* Agnello, M. Carbone & P. Alvarado and *O. alutacea* (Pers.) Massee is similar to that of *O. bomiensis*, but *O. adorniae* differs from *O. bomiensis* in having smaller ascospores (10.5–)11–12(–12.5) × 6–6.5(–7) µm, *O. alutacea* differs in darker receptacle surface color, which is yellowish brown, sometimes with purplish brown tones [8,10]. *Otidea aspera* also has pale yellow receptacle surface similar to *O. bomiensis*, but *O. aspera* differs in having shorter ascospores (12–)12.8–15(–15.5) × (5.8)6.5–7.5(–8) µm and high warts 50–80 µm on the receptacle surface [15].

***Otidea******gongnaisi******ensis*** L. Fan & Y.Y. Xu, sp. nov. (Figure 5)

MycoBank: MB844137

Etymology: *gongnaisi**ensis*, referring to the locality where the type specimen was collected.

Holotype: China. Xinjiang Autonomous Region, Gongnaisi National Forest Park, August 1994, J.Y. Wang, 132 (HMAS 69951).

Saprobic on cone or soil. Apothecia solitary or gregarious, 15–40 mm high, 10–25 mm wide, long ear-shaped or broadly ear-shaped, split, stipitate. Hymenium yellowish beige when fresh, dirty yellow with pale brown tones when dry, subsmooth. Receptacle surface concolorous with hymenium when fresh, slightly hygrophanous, dirty yellow when dry, furfuraceous. Stipe 5–15 × 3–7 mm. Basal tomentum and mycelium whitish. Apothecial section 700–1000 µm thick. Ectal excipulum of *textura angularis*, 75–150 µm thick, cells thin-walled, brownish, 12–30 × 10–26 µm. Medullary excipulum of *textura intricata*, 300–500 µm thick, hyphae 3–9 µm wide, thin-walled, septate, hyaline to light brown. Subhymenium c. 75–120 µm thick, visible as a yellowish brown zone of densely arranged cylindrical to swollen cells. Paraphyses septate, curved to hooked of uniform width or slightly enlarged at the apices to 2.5–4 (5) µm wide, without or with 1–2 low notches, sometimes with 1–2 slightly swollen areas near the apex. Asci 150–200 × 8–12 µm, 8-spored, unitunicate, operculate, cylindrical, hyaline, non-amyloid, long pedicellate, arising from croziers, ascospores release from an eccentric split at the apical apex. Ascospores overlapping uniseriate, ellipsoid, hyaline, with one to two large guttules, smooth, (10.2–)10.6–12.2(–12.5) × (5.7–)6–6.8(–7.1) µm (L_m_ × W_m_ = 11.3 × 6.4 µm, Q = 1.6–1.9, Q_m_ = 1.8, n = 50). Receptacle surface almost seldom warts, sometimes with scattered hyphoid hairs 20–30 µm high, of 2–3 subglobose to elongated cells, slightly constricted at septa, 6–11 µm wide. Resinous exudates present on the receptacle surface, brown to dark brown, dissolving into amber drops in MLZ, partially dissolving and turning yellowish brown in KOH. Basal mycelium of 3–5 µm wide, septate, hyaline to pale brown hyphae, unchanged in KOH, with many, small, irregularly, brown, resinous exudates on the surface, dissolving in MLZ, partially and more slowly in KOH.

Additional specimens examined: China. Xinjiang Autonomous Region, Jimusa’er, alt. 1700 m, on cone of *Picea* sp., 2 August 2003, W.Y. Zhuang & Y. Nong, 4670 (HMAS 83574).

Notes: *Otidea*
*gongnaisiensis* is characterized by the stipitate, yellowish beige, broadly ear-shaped apothecia, seldom warts on receptacle surface, and excipular resinous exudates partially dissolving and turning yellowish brown in KOH. *Otidea formicarum* Harmaja and *O. pseudoformicarum* A.H. Ekanayaka, Q. Zhao & K.D. Hyde are similar in apothecial shape and color to *O. gongnaisiensis*, but *O. formicarum* can be distinguished by its relatively small, reddish brown apothecia, comparatively shorter ascospores (L_m_ = 10–10.7 µm), and unique habitat, often occurring on anthills [8]. *Otidea pseudoformicarum* differs in shorter asci (115–150 × 7–10 µm) and smaller ascospores (8–10 × 5–7 µm) [13]. The ascospores size of *O. subformicarum* Olariaga, Van Vooren, Carbone & K. Hansen is similar to *O*. *gongnaisiensis*, but it can be distinguished by orange brown to reddish brown apothecial color, and high warts (45–65 µm) on the receptacle surface [8]. *Otidea shennongjiana* can also be distinguished from *O. gongnaisiensis* by orange tones of apothecia, smooth basal mycelium, and the resinous exudate in the ectal excipulum partly dissolving into drops in KOH.

***Otidea******hanzhongen******sis*** L. Fan, M.Q. Zhu & Y.Y. Xu, sp. nov. (Figure 6)

MycoBank: MB844138

Etymology: *hanzhongensis*, referring to the locality where the type specimen was collected.

Holotype: China. Shaanxi Province, Hanzhong City, Zuoxi River Management Station, alt. 1200 m, 2 October 2016, M.Q. Zhu, (610723MF0034).

Saprobic on soil. Apothecia solitary or gregarious, 10–40 mm high, 20–50 mm wide, broadly ear-shaped to shallowly cup-shaped or shallowly disc, broader above, split, sessile or shortly stipitate. Hymenium pale greyish yellow to pale ochre when fresh, dark yellowish brown to brown when dry, subsmooth. Receptacle surface pale yellow when fresh, slightly hygrophanous, yellowish brown when dry, furfuraceous to finely warty. If present, the stipe is very short. Basal tomentum and mycelium whitish to grayish white. Apothecial section 600–850 µm thick. Ectal excipulum of *textura angularis*, 80–130 µm thick, cells thin-walled, pale brown, 11–30 × 9–22 µm. Medullary excipulum of *textura intricata*, 250–450 µm thick, hyphae 3–8.5 µm wide, thin-walled, septate, hyaline to light brown. Subhymenium ca. 50–100 µm thick, visible as a yellowish brown zone of densely arranged cylindrical to swollen cells. Paraphyses septate, curved to hooked, usually enlarged at the apices, 3.5–5 μm wide at apex, 2–2.5 μm below, without notch. Asci 120–160 × 8.5–11.5 µm, 8-spored, unitunicate, operculate, cylindrical, hyaline, non-amyloid, long pedicellate, arising from croziers, ascospores released from an eccentric split at the apical apex. Ascospores overlapping uniseriate, ellipsoid, hyaline, with one to two large guttules, smooth, (10.5–)11–13(–13.5) × (5.5–)6–6.8(–7) µm (L_m_ × W_m_ = 12 × 6.5 µm, Q = 1.7–2, Q_m_ = 1.85, n = 50). Receptacle surface with broadly conical warts, 40–60 µm high, formed by hyphoid hairs, of 3–5 subglobose to elongated cells, constricted at septa, 5–9 µm wide. Resinous exudates absent to scarce. Basal mycelium of 2.5–5 µm wide, septate, hyaline to pale brown hyphae, unchanged in KOH, smooth.

Notes: *Otidea hanzhongensis* is characterized by the pale greyish yellow to pale ochre hymenium, pale yellow receptacle color, small ascospores, short asci, and the lack of resinous exudates on the ectal excipulum and basal mycelium. *Otidea parvispora* (Parslow & Spooner) M. Carbone, Agnello, Kautmanová, Z.W. Ge & P. Alvarado and *O. aspera* also share similar light-colored apothecia, but *O. parvispora* can be distinguished by light ochraceous-buff hymenium, pale fawn receptacle surface, and longer asci 165–185 × 9–10 µm [11]. *Otidea aspera* differs from having longer ascospores (12–)12.8–15(–15.5) × (5.8–)6.5–7.5(–8) µm and longer asci 150–200 × 9–13 µm [15]. In the *O. alutacea* clade, the other new collected species, *O. bomiensis,* is also easily distinguished from *O. hanzhongensis* by the light brown to brown hymenium, bigger ascospores (14.5–)15–16.8(–17.2) × (6.5–)7.1–7.9(–8.3) µm and longer asci 150–200 × 10–14 µm.

***Otidea******shennongjiana*** L. Fan & Y.Y. Xu, sp. nov. (Figure 7)

MycoBank: MB844139

Etymology: *shennongjiana*, referring to the locality where the type specimen was collected.

Holotype: China. Hubei Province, Shennongjia National Forest Park, changyanwu, 29 July 1984, J.X. Tian, 75 (HMAS 53691).

Saprobic on rotten wood and roots. Apothecia gregarious to caespitose, 8–38 mm high, 5–15 mm wide, broadly ear-shaped, sometimes cup-shaped, split, stipitate. Hymenium light yellow with orange tones to pale orange when fresh, orange brown to reddish brown when dry, subsmooth. Receptacle surface pale yellow when fresh, slightly hygrophanous, pale yellowish brown, margin pale reddish brown when dry, furfuraceous to finely warty. Stipe 5–15 × 3–5 mm. Basal tomentum and mycelium white. Apothecial section 550–900 µm thick. Ectal excipulum of *textura angularis*, 80–120 µm thick, cells thin-walled, hyaline to brown, 11–26 × 9–16 µm. Medullary excipulum of *textura intricata*, 200–350 µm thick, hyphae 3–8 µm wide, thin to slightly thick walled, hyaline to light brown. Subhymenium c. 75–150 µm thick, visible as a brown zone, of densely arranged cylindrical to swollen cells. Paraphyses septate, straight to curved, a few hooked, of uniform width at the apices to 2.2–3.5 µm wide, without or a few with 1–2 low notches. Asci 135–175 × 8–12 µm, 8-spored, unitunicate, operculate, cylindrical, hyaline, non-amyloid, long pedicellate, arising from croziers, ascospores released from an eccentric split at the apical apex. Ascospores overlapping uniseriate, ellipsoid, hyaline, with one to two large guttules, smooth, (10.5–)11–13(–13.5) × (5.8–)6.2–7.2(–7.5) µm (L_m_ × W_m_ = 12 × 6.8 µm, Q = 1.6–1.9, Q_m_ = 1.76, n = 50). Receptacle surface with warts, 30–50 µm high, formed by short, fasciculate, hyphoid hairs, of 3–6 subglobose to elongated cells, constricted at septa, 5–10 µm wide. Resinous exudates abundant on the receptacle surface, yellow brown to dark brown, dissolving into amber drops in MLZ, and partly dissolving into amber drops in KOH. Basal mycelium of 2.5–5.5 µm wide, septate, hyaline to pale brown hyphae, unchanged in KOH, smooth.

Additional specimens examined: China. Hubei Province, Shennongjia National Forest Park, changyanwu, 4 August 1984, J.X. Tian, 95 (HMAS 53692).

Notes: *Otidea shennongjiana* is characterized by the broadly ear-shaped apothecia, light yellow with orange tones hymenium, pale yellow receptacle surface, big ascospores, smooth basal mycelium, and by its habitat, often occurring on rotten wood and roots. Same as with *O. shennongjiana*, the apothecia of *O. nannfeldtii* Harmaja, and *O. subformicarum* all have orange tones, but *O. nannfeldtii* differs from *O. shennongjiana* by the smaller ascospores (9–)9.5–10.5(–11.5) × 5.5–6.5(–7) µm, excipular resinous exudates turning reddish brown in KOH and with yellow resinous exudates on basal mycelium [8]. *Otidea subformicarum* differs in narrow ascospores (10.5–12 × 6–6.5 µm), long asci (184–237 × 11–11.5 µm), and pale yellow drops on basal mycelium surface [8]. In the *O. formicarun* clade, *Otidea khakicolorata* L. Fan & Y.Y. Xu also have smooth basal mycelium, but it can be distinguished by khaki to pale ochre apothecia, smaller ascospores (8.5–)9–10(–10.5) × 4.5(5)–6(–6.5) µm, and excipular resinous exudates turning reddish brown in KOH [15]. Phylogenetic analyses revealed that *O. shennongjiana* was grouped with an undescribed species *Otidea* sp. ‘b’, with high support values (Figure 1 and Figure 2). Mean ascospores length and width of *Otidea* sp. ‘b’ is 11.1 × 6.7 µm, which is smaller than *O. shennongjiana*. DNA analysis showed that they shared only 95.45% ITS sequence similarity, suggesting that *O. shennongjiana* and *Otidea* sp. ‘b’ may be different species.

***Otidea aspera*** L. Fan & Y.Y. Xu, Journal of Fungi 8 (3, no. 272): 9 (2022) (Figure 8E,F)

Specimens examined: China, Shanxi Province, Pangquangou National Nature Reserve, Badaogou, alt. 2000 m, 28 August 2018, J.Z. Cao, LH251 (HSA 251).

Notes: This species has been recently described in Shanxi Province and Inner Mongolia, China [15]. The specimen (HSA 251) collected from Shanxi Province was confirmed to be *O. aspera* by morphological and molecular evidence in this study. As shown in Figure 8E, the color of the hymenium of HSA 251 is grayish brown to brown, which is darker than the original descriptions (greyish yellow to light brown) of *O. aspera*. This may be due to the different maturity stages at which different fruit-bodies were collected. Therefore, the color of the hymenium of *O. aspera* was re-described as follows: hymenium surface grayish yellow, light brown to brown when fresh.

***Otidea sinensis*** J.Z. Cao & L. Fan, Mycologia 82(6): 736 (1990) (Figure 8C)

Specimens examined: China, Sichuan Province, Xiaojin County, National Scenic Area of Four Girls Mountain, alt. 2014 m, 15 August 2013, W.L. Lu, 1734 (HMAS 268390); ibid., 15 August 2013, W.L. Lu, 1728 (HMAS 268544).

Saprobic on soil. Apothecia gregarious to caespitose, 10–30 mm high, 5–25 mm wide, long ear-shaped, or broadly ear-shaped, split, stipitate. Hymenium cream to greyish yellow when dry, subsmooth. Receptacle surface pale yellowish brown when dry, finely warty. Stipe 5–10 × 3–6 mm. Basal tomentum and mycelium white. Apothecial section 600–1000 µm thick. Ectal excipulum of textura angularis, 100–150 µm thick, cells thin-walled, hyaline to brown, 12–35 × 10–24 µm. Medullary excipulum of textura intricata, 400–500 µm thick, hyphae 3–9 µm wide, thin-walled, hyaline. Subhymenium c. 50–80 µm thick, visible as a pale brown zone, of densely arranged cylindrical to swollen cells. Paraphyses septate, curved to hooked, slightly enlarged at the apices to 3–5.5 µm wide, without or a few with 1–2 low notches. Asci 150–180 × 8–11 µm, 8-spored, unitunicate, operculate, cylindrical, hyaline, non-amyloid, long pedicellate, arising from croziers, ascospores released from an eccentric split at the apical apex. Ascospores overlapping uniseriate, ellipsoid to slightly fusoid, hyaline, with one to two large guttules, smooth, 10.5–13 × 5.5–6.7 µm (L_m_ × W_m_ = 11.8 × 6.3 µm, Q = 1.7–2, Q_m_ = 1.88, n = 50). Receptacle surface with warts, 50–85 µm high, formed by short, fasciculate, hyphoid hairs, of 4–6 subglobose to elongated cells, constricted at septa, 4–9 µm wide. Resinous exudates abundant on the receptacle surface, yellow to yellow brown, partially dissolving and turning slightly reddish in MLZ, unchanged in KOH. Basal mycelium of 3–7 µm wide, septate, hyaline to pale brown hyphae, turning yellow in KOH, with small, irregularly, brown, resinous exudates on the surface, partially dissolving in KOH, partially dissolving in MLZ.

Notes: The two specimens (HMAS 268390 and HMAS 268544) grouped together with the type specimen of *O. sinensis* with strong a support value (BS = 91%, PP = 1.00). We examined the morphology of these two specimens, which are consistent with the original description by Cao et al. [31]. The two specimens were, therefore, confirmed to be *O. sinensis*. The type specimen of *O. sinensis* was collected from Heilongjiang Province in northeast China. Our results demonstrate that *O. sinensis* is also distributed in southwest China.

## 4. Discussion of Additional Specimens

In the phylogeny derived from the combined datasets (LSU, *tef1-α*, and *rpb2*) (Figure 1), a specimen (HMAS 85660) collected from Heilongjiang Province, China, clustered in the *O. bufonia-onotica* clade, and formed a sister clade with *O. cupulata* L. Fan & Y.Y. Xu with high support values. This specimen collected from rotted wood has an ascospore size of (13.5–)14–15.7(–16) × (6.3–)6.6–7.7(–8) µm, Q = 1.9–2.2, which overlaps with *O. cupulata*. However, the ascospore shape of the two species is different. The ascospores in HMAS 85660 is narrowly fusoid, while ellipsoid to slightly subfusoid in *O. cupulata*. Furthermore, HMAS 85660 had less than 95.28% ITS similarity with *O. cupulata*. Therefore, HMAS 85660 may represent a new species, but due to the poor condition of this specimen, its formal taxonomic treatment awaits the discovery of new fruitbodies.

A specimen (HMAS 23948) identified as *O. olivaceobrunnea* Harmaja by Xu et al. [14] was inserted into the clade of *O. purureogrisea* Pfister, F. Xu & Z.W. Ge. We borrowed HMAS 23948 and the paratype of *O. purureogrisea* (WZ 2157, i.e., HMAS 72805 = FH 00464724) and performed morphological examinations. Although Xu et al. [14] described the ascospores of *O. purureogrisea* as ellipsoid, it can be seen from the original description Figure 6B that its ascospores are fusoid (narrowed at both ends) rather than ellipsoid. Our result also shows that the ascospores of HMAS 72805 is narrow fusoid (Figure 8A), and its size is 15–18 × 6.5–8.2 µm, slightly larger than 11–16 × 6–7.5 µm originally reported by Xu et al. [14]. Morphological examination of HMAS 23948 shows that the ascospores are fusoid (Figure 8B) and 15.5–17.5 × 6.5–8 µm in size, which is consistent with HMAS 72805. Moreover, according to Cao et al. [31] and Zhuang [32], *O*. *olivaceobrunnea* has ellipsoid ascospores, and olive-brown hymenium when fresh, becoming blackish brown to black when dry. We, therefore, believe that HMAS 23948 is not *O. olivaceobrunnea* but maybe *O. purureogrisea*.

In the *O. alutacea* clade, *Otidea bomiensis* forms a sister clade with clade 6 (containing four Chinese collections) with a moderate support value (BS = 73%, PP = 1.00) (Figure 1). The four specimens in clade 6 are of poor quality due to the early collection time, and the macroscopic features can hardly be observed, but we checked their microscopic features (Figure 8D). The ascospore measurements we obtained were 15–20 × 6–8.5 µm (Q = 2–2.5, Q_m_ = 2.2), which is somewhat longer than that described by Xu et al. [14] (14–17.5 × 6–8 µm, Q_m_ = 2). Therefore, *Otidea bomiensis* differs from clade 6 by having shorter ascospores and a smaller Q_m_ value. Furthermore, in the phylogenetic tree derived from the *O. alutacea* dataset (Figure 3), *O. bomiensis* and clade 6 form two distant clades, indicating that they are two distinct species.

Clade 7 contains two specimens, the Swedish C-F-48045 is without any published morphological data, and the other HMAS 88262 is from the Xinjiang Autonomous Region of China. *Otidea hanzhongensis* forms a sister clade with clade 7 with good support values (BS = 73%, PP = 1.00) (Figure 1), but HMAS 88262 differs from *O. hanzhongensis* (ascospore length < 13.5 µm, Q = 1.7–2) in having significantly larger ascospores (16.5–18.5 × 7.4–8.2 µm, Q = 2.1–2.4). The phylogenetic tree derived from the *O. alutacea* dataset (Figure 3) also confirmed that they are two different species. Xu et al. [14] treated C-F-48045 as a separate clade (clade 7); our collections (HMAS 88262) fall into this clade, but here we did not treat this taxon taxonomically, as we could not study the Swedish collection, and our collection (HMAS 88262) is in poor condition.

Regarding the species represented by *Otidea* sp. ‘c’, clades 6 and 7, we still need to go to the original collection site to collect new specimens for further research. In addition, many sequences of the genus *Otidea* obtained from ectomycorrhizal, and soil samples have been submitted into public databases without confirmed species identification. These should be considered in future research to extend and improve our knowledge of the biogeography and diversification of *Otidea* species worldwide.

## 5. Conclusions

In this study, 11 *Otidea* specimens deposited in Chinese herbaria and one newly collected specimen from northern China were examined based on morphological and phylogenetic analyses. The results revealed a total of nine phylogenetic species, of which four were described as new, namely *O. bomiensis*, *O. gongnaisiensis*, *O. hanzhongensis*, and *O. shennongjiana*. In addition, two known species were identified, namely *O. aspera* and *O. sinensis*. The remaining three taxa are putative new species, but more specimens need to be collected for further study before attempting a formal taxonomic treatment. Our findings emphasize that the diversity of the *Otidea* species in China is extremely high and that more studies are needed to fully appreciate the exact species number.

## Figures and Tables

**Figure 1 biology-11-00866-f001:**
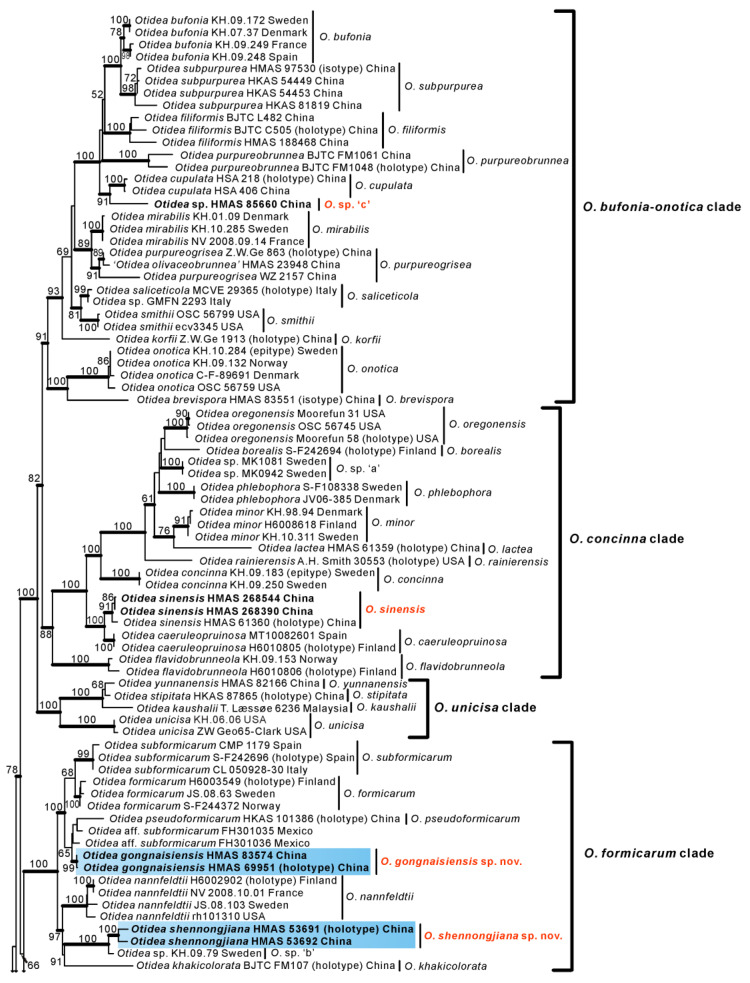
Phylogeny of *Otidea* derived from maximum likelihood analyses of the combined (LSU-*tef1-α*-*rpb2)* dataset. *Monascella botryosa* and *Warcupia terrestris* were used to root the tree as outgroups. Maximum likelihood bootstrap support (BS ≥ 50%) is shown on the nodes. Branches with Bayesian posterior probabilities (PP) ≥ 0.95 and new sequences are in bold.

**Figure 2 biology-11-00866-f002:**
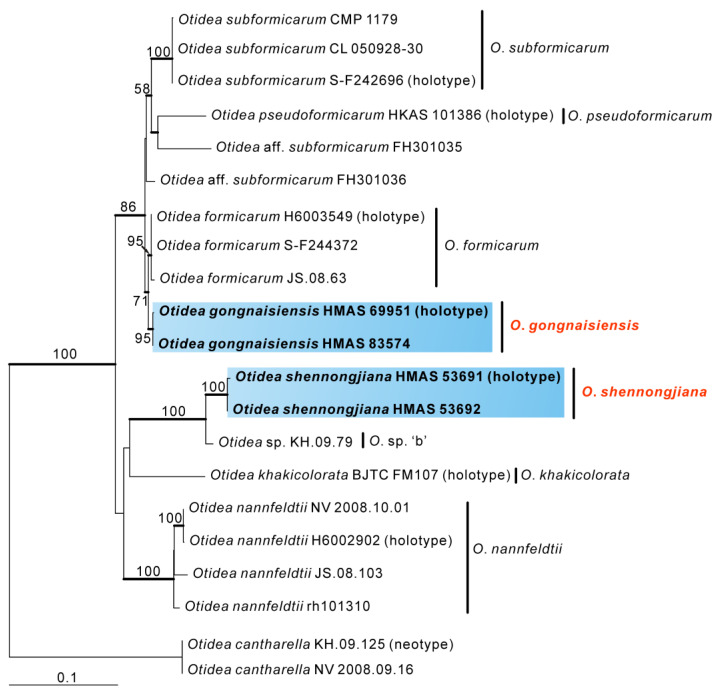
Phylogeny of the *O**. formicarum* clade derived from maximum likelihood analyses of the combined (ITS-LSU) dataset. *Otidea cantharella* was used to root the tree as outgroups. Maximum likelihood bootstrap support (BS ≥ 50%) is shown on the nodes. Branches with Bayesian posterior probabilities (PP) ≥ 0.95 and new species are in bold.

**Figure 3 biology-11-00866-f003:**
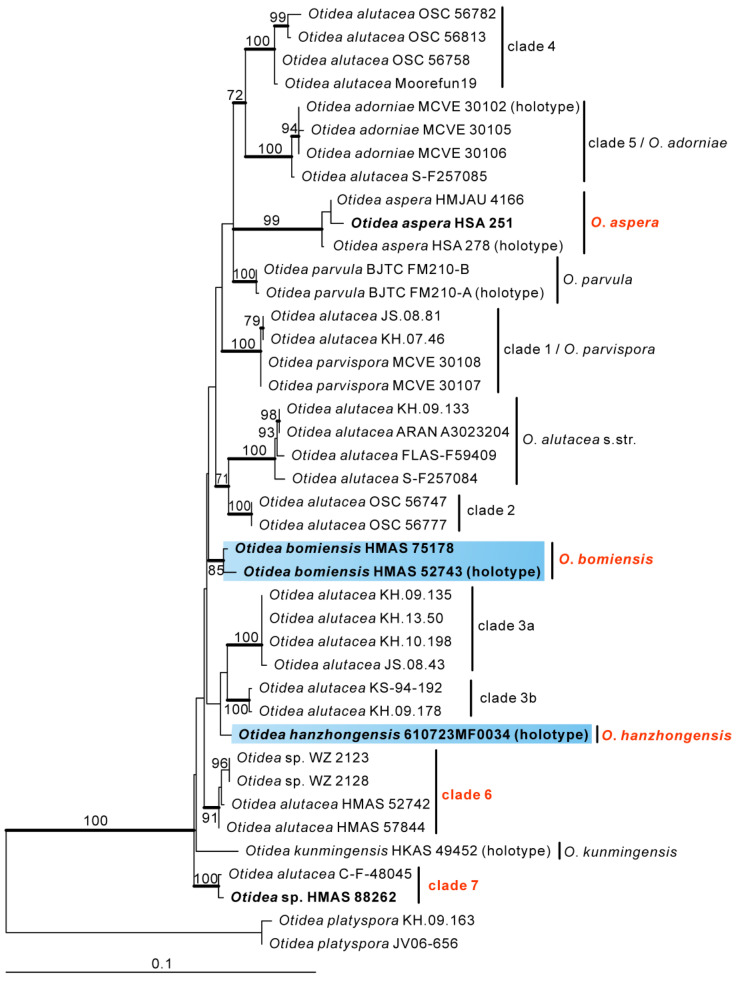
Phylogeny of the *O**. alutacea* clade derived from maximum likelihood analyses of the combined (ITS-LSU) dataset. *Otidea platyspora* was used to root the tree as the outgroups. Maximum likelihood bootstrap support (BS ≥ 50%) is shown on the nodes. Branches with Bayesian posterior probabilities (PP) ≥ 0.95 and new sequences are in bold.

**Figure 4 biology-11-00866-f004:**
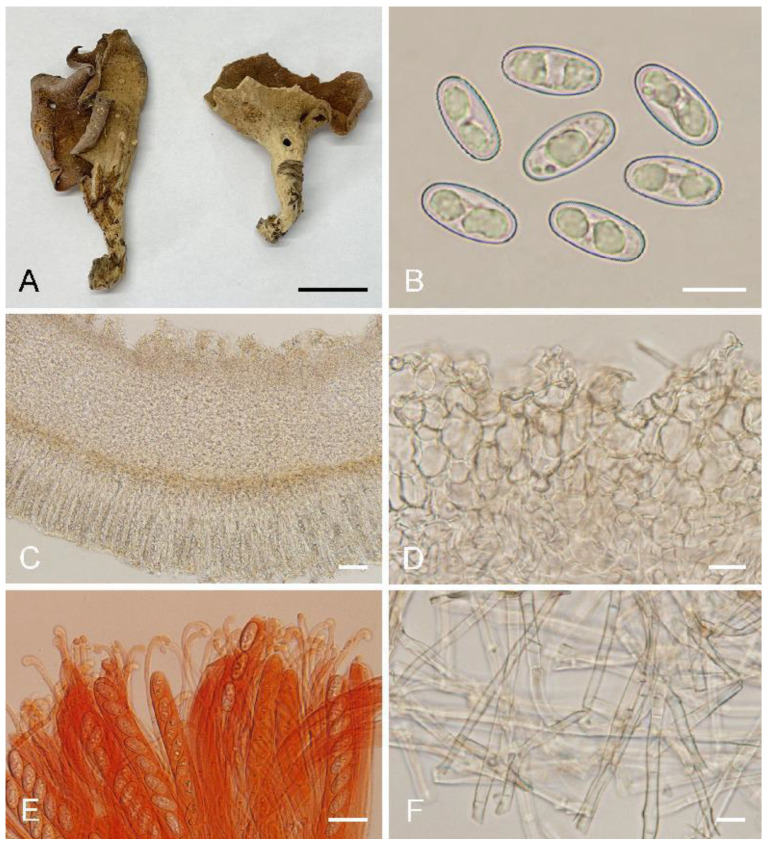
*Otidea**bomiensis* (HMAS 52743) (**A**) apothecia, (**B**) ascospores, (**C**) anatomy of apothecium, (**D**) ectal excipulum in water, (**E**) asci and paraphyses in Congo Red, and (**F**) basal mycelium. Scale bars: (**A**) = 1 cm, (**B**,**F**) = 10 μm, (**C**) = 100 μm, (**D**,**E**) = 20 μm.

**Figure 5 biology-11-00866-f005:**
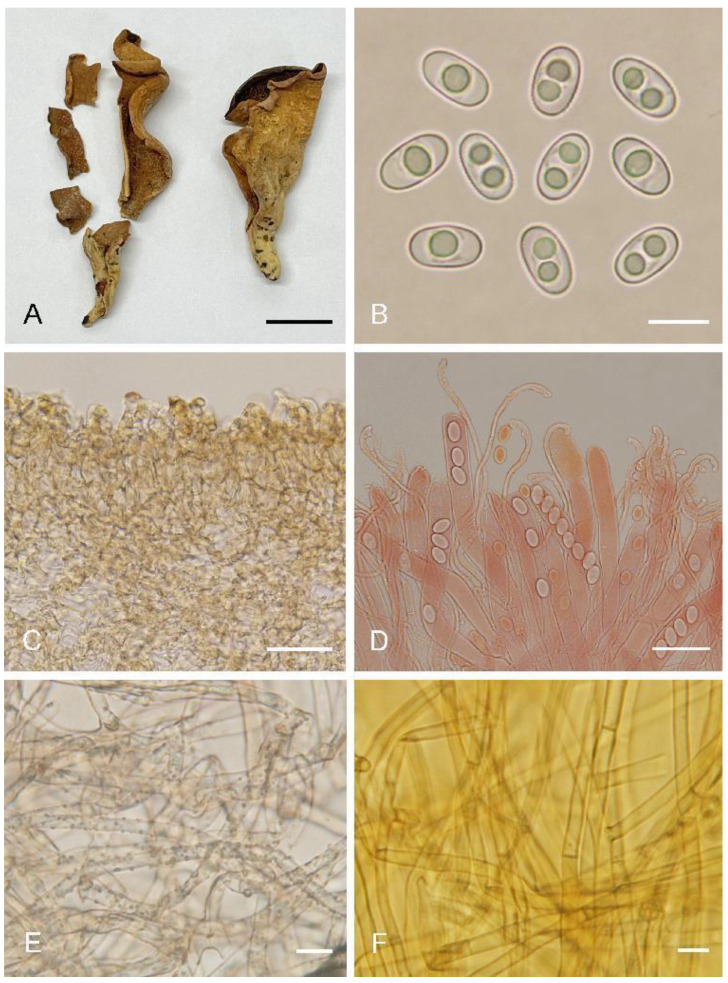
*Otidea**gongnaisiensis* (HMAS 69951) (**A**) apothecia, (**B**) ascospores, (**C**) ectal and medullary excipulum in KOH, (**D**) asci and paraphyses in Congo Red, (**E**) basal mycelium in water, and (**F**) basal mycelium in MLZ. Scale bars: (**A**) = 1 cm, (**B**,**E**,**F**) = 10 μm, (**C**) = 50 μm, (**D**) = 30 μm.

**Figure 6 biology-11-00866-f006:**
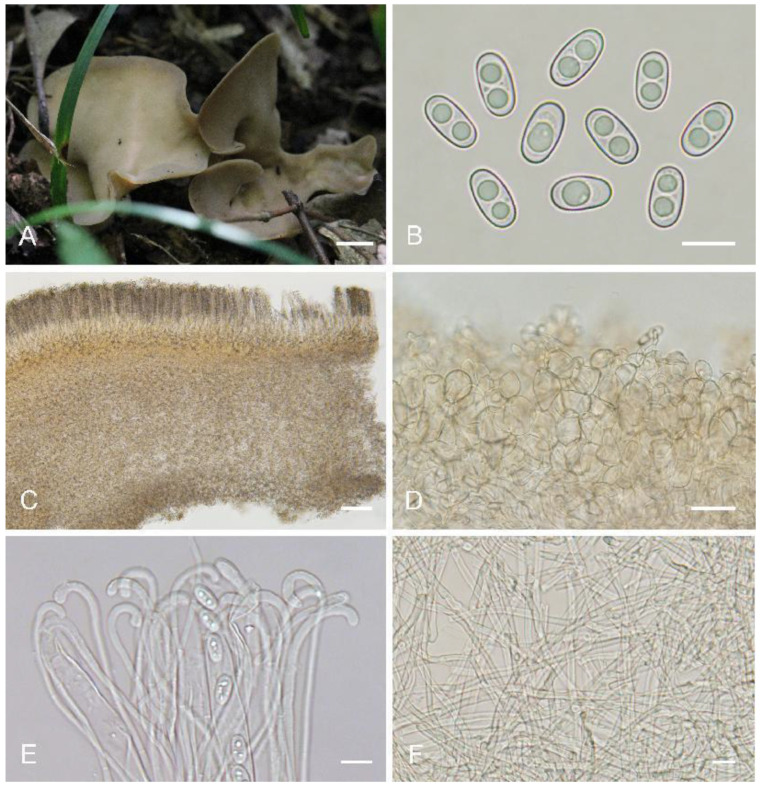
*Otidea**hanzhongensis* (610723MF0034) (**A**) apothecia, (**B**) ascospores, (**C**) anatomy of apothecium, (**D**) ectal excipulum in water, (**E**) paraphyses, and (**F**) basal mycelium. Scale bars: (**A**) = 1 cm, (**B**,**E**,**F**) = 10 μm, (**C**) = 100 μm, (**D**) = 30 μm.

**Figure 7 biology-11-00866-f007:**
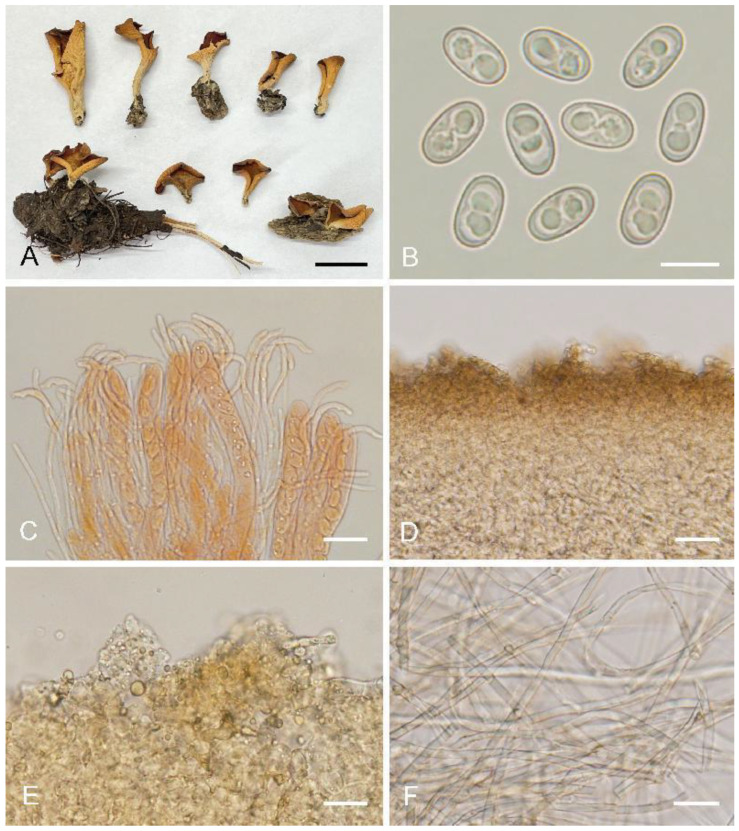
*Otidea shennongjiana* (HMAS 53691) (**A**) apothecia, (**B**) ascospores, (**C**) paraphyses and asci in Congo Red, (**D**) ectal and medullary excipulum in water, (**E**) amber drops on the outermost ectal excipulum cells in KOH, and (**F**) basal mycelium. Scale bars: (**A**) = 1 cm, (**B**) = 10 μm, (**C**,**E**,**F**) = 20 μm, (**D**) = 50 μm.

**Figure 8 biology-11-00866-f008:**
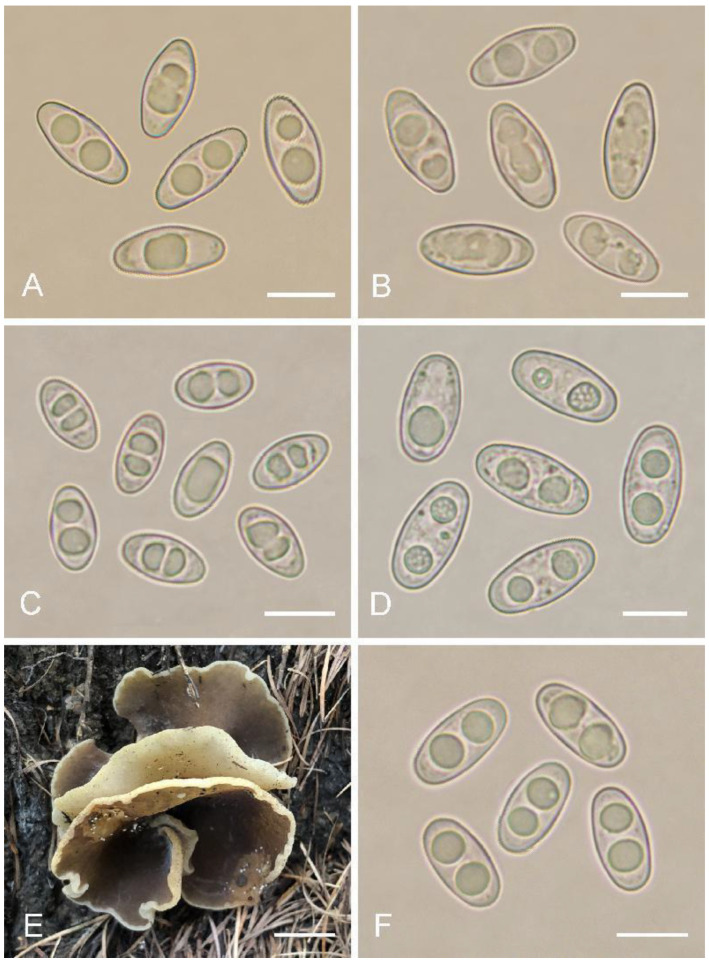
(**A**–**D**,**F**) ascospores, (**A**) *Otidea purureogrisea* WZ 2157 (HMAS 72805), (**B**) *Otidea* cf. *purureogrisea* (HMAS 23948), (**C**) *Otidea sinensis* (HMAS 268544), (**D**) *Otidea* sp. WZ 2123 (HMAS 72058), (**F**) *Otidea aspera* (HSA 251), and (**E**) apothecia (HSA 251). Scale bars: (**A**–**D**,**F**) = 10 μm, (**E**) = 1 cm.

## Data Availability

The sequencing data were submitted to GenBank.

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
