# Peer review of "Phylogenetic Analysis Reveals Four New Species of Otidea from China"

_biology, 2022, doi:10.3390/biology11060866_

Round 1

Reviewer 1 Report

Dear Authors

The description of four new species of Otidea from China using morphological and molecular data, presented by the authors seems interesting to me. The manuscript overall is good. The content of the paper could be published in Biology. In the manuscript, I pointed out a few minor corrections.

Best regards

Reviewer 3 Report

Dear Authors,

Please find herewith my review comments on the manuscript. However, some grammar errors must be corrected. Additionally, please correct the reference style in line with the journal guidelines.

Kind regards

Round 2

Reviewer 2 Report

I accept the changes that have been made in accordance with my previous suggestions.

Author Response

We appreciate the insightful comments by the anonymous reviewer, which greatly improved the manuscript.

Reviewer 3 Report

Dear authors,

I reviewed this manuscript before and most of my suggestions are considered. However, there are a few issues I would like to be addressed in this version. 

Thank you

Author Response

Dear reviewer,

Thank you for your comments and constructive suggestions concerning our manuscript. We have made corrections that we hope meet with approval. All the corrections and changes are marked in blue in the revised manuscript.

Thank you.